# Biocontrol and Probiotic Function of Non-*Saccharomyces* Yeasts: New Insights in Agri-Food Industry

**DOI:** 10.3390/microorganisms11061450

**Published:** 2023-05-30

**Authors:** Francesca Comitini, Laura Canonico, Alice Agarbati, Maurizio Ciani

**Affiliations:** Dipartimento di Scienze della Vita e dell’Ambiente (DiSVA), Università Politecnica delle Marche, Via Brecce Bianche, 60131 Ancona, Italy; l.canonico@univpm.it (L.C.); a.agarbati@univpm.it (A.A.)

**Keywords:** biocontrol, non-*Saccharomyces* yeasts, functional yeasts, probiotics

## Abstract

Fermented food matrices, including beverages, can be defined as the result of the activity of complex microbial ecosystems where different microorganisms interact according to different biotic and abiotic factors. Certainly, in industrial production, the technological processes aim to control the fermentation to place safe foods on the market. Therefore, if food safety is the essential prerogative, consumers are increasingly oriented towards a healthy and conscious diet driving the production and consequently the applied research towards natural processes. In this regard, the aim to guarantee the safety, quality and diversity of products should be reached limiting or avoiding the addition of antimicrobials or synthetic additives using the biological approach. In this paper, the recent re-evaluation of non-*Saccharomyces* yeasts (NSYs) has been reviewed in terms of bio-protectant and biocontrol activity with a particular focus on their antimicrobial power using different application modalities including biopackaging, probiotic features and promoting functional aspects. In this review, the authors underline the contribution of NSYs in the food production chain and their role in the technological and fermentative features for their practical and useful use as a biocontrol agent in food preparations.

## 1. Introduction

Agri-food systems encompass the primary production of food products, as well as food storage, aggregation, post-harvest handling, transportation, processing, distribution, marketing, disposal and consumption. Microorganisms and their metabolites can support technologies to produce more sustainable products at different stages of the food chain. Traditional industrial fermentation attributes to *Saccharomyces cerevisiae* the most important role as a biotechnological organism involved in worldwide fermentation products such as beers, cider, wines, sake, distilled spirits, bakery products, cheese, sausages and other fermented foods. However, in the last 20 years, the world of research, and in parallel the industrial one, has started to re-evaluate the potential positive contribution of non-*Saccharomyces* yeasts (NSYs). They have found a pro-technological use in traditional fermentations, where they can impart peculiar and distinctive characteristics to the product, but also in other applications such as in biomedical or fundamental biological research, environmental biotechnology, heterologous protein production, biocontrol and food and feed sectors [1]. The utility of NSYs in the food field is accelerating due to a wide number of metabolic abilities that confer aroma and structure to the final products but also due to their plasticity in assimilating different substrates. They generally show low fermentation yields and higher sensitivity to ethanol stress if compared with *S. cerevisiae*. On the other hand, NSYs may display a great range of fermentation metabolites and end products by providing distinctive features [2].

In any case, the positive or negative contribution that each non-*Saccharomyces* species gives to the product strictly depends on the matrix and fermentation context (Figure 1).

Until a few decades ago, in traditional fermented beverages, NSYs were considered synonymous with spoilage yeasts. Differently, today many researchers clearly highlight their potential for species-specific or strain-specific traits, and it is unequivocally accepted that NSYs represent an important resource with economic repercussions in the market of fermented products. In the worldwide market, there is a growing interest in “non-conventional” yeast strains, another way to define NSYs, that can help generate the diversity and the complexity desired in diversified and aware consumers. Effectively, selected strains of NSYs contribute to set up innovative fermentations in the food and beverages field, starting with winemaking and continuing in brewing where they impart specific “bioflavoring” or functionalize the beverages and reduce the final ethanol content. In addition, NSYs may be used as alternative starters to promote biodiversity and quality of other different fermented foods such as bakery products. In a recent paper, some selected non-*Saccharomyces* wine strains were assayed in the production of leavened doughs [3]. The use of several NSYs species, such as *Hanseniaspora uvarum*, *Metschnikowia pulcherrima*, *Pichia kudriavzevii*, *Torulaspora delbruekii* and *Zygotorulaspora florentina,* were determined in the bakery industry as an enhancement of diversity and may be useful in reducing or avoiding yeast intolerance.

Some NSYs including *Kluyveromyces lactis* have been able to survive in conditions that mimic the gastrointestinal environment and can form biofilms on an abiotic medium such as polystyrene [3]. These yeast strains also exhibited highly hydrophilic cell wall surface properties and adhesion ability to intestinal Caco-2 cells, thus increasing their potential as probiotic strains. Effectively, in the pharmacological industry, the rational design of probiotics includes, in addition to their health claims, several other criteria: stability during manufacturing processes, viability during gastrointestinal transit and functionality at the desired target site. Host safety and the management of their formulation and viability are also determining characteristics in the selection of probiotics. In relation to these multiple traits, the last decade has seen much evidence regarding NSYs as probiotic candidates. For example, a report by Li and colleagues [4] clarified the GRAS status of several *Kluyveromyces* species. In a general picture, yeast may offer more benefits than bacteria, as they are insensitive to antibiotics and can be easily used for the treatment of antibiotic-associated diarrhea.

The probiotic and biocontrol features as well as the general interest that revolves around the characteristics and the use of NSYs in the agri-food industry are the concerns of the present review. In this regard, an updated survey of the main recent publications focusing on this topic with scientific and market implications is prepared. The antimicrobial and probiotic attitudes are analyzed with particular attention to pointing out the recent findings, indicating the perspectives and emphasizing the fields of interest that are still poorly investigated.

## 2. Antimicrobial Activity of Non-*Saccharomyces* Yeasts (NSYs)

Already more than twenty years ago some authors [5,6] anticipated the potential use of NSYs during the first step of fermentation applied to improve the final flavor of wines. Today, knowledge has exponentially increased and the aromatic enhancement of wines due to the use of NSYs is only one of the many other features identified for their potential application. In view of the possible use in agri-food industry, it is a common prerogative that NSYs must possess specific traits to be selected. Indeed, after the general re-valuation of the role of NSYs, several studies have focused the attention on their multiple advantages, particularly in winemaking. Among these, the antagonistic activity against undesired microorganisms is of paramount importance.

It has been widely demonstrated that some NSY strains in winemaking can control spoilage yeasts or filamentous fungi both in the vineyard and during the early stages of fermentation [7,8,9]. In this regard, spoilage species producing off-flavors or films in bulk wine such as *Zygosaccharomyces rouxii*, *Kloeckera apiculata, Pichia* spp. and *Candida* spp. may be prevented by the use NSYs through released active antimicrobial extracellular molecules [10]. Effectively, the growing interest in reducing and rationalizing the use of SO_2_ in winemaking has urged scientific investigations towards the antimicrobial activity of NSYs as an alternative to conventional chemical additives [11,12]. About this, natural control strategies may involve the use of killer toxins (mycocins), antimicrobial peptides such as Lactoferricin B or volatile compounds produced by NSYs as a biological strategy to counteract contamination [13].

### 2.1. Biological Control

The term “biocontrol” related to the use of microorganisms as natural biological agents was defined as the reduction in pathogen or disease activities through organisms or their molecules. In agri-food, this concept is related to an alternative strategy to the use of chemical products and the use of microorganisms with antagonist action against other microorganisms lowering the use of pesticides and boosting food quality and safety [14,15,16].

In recent years, to avoid the huge losses of fruit and vegetables due to pathogens, biological strategies have been studied as alternatives to products of a chemical nature. The addition of microorganisms as bio-protective agents or their antimicrobial products has already been identified as “bio-protection”. This practice is firstly used in agriculture and then in food industries for the control of fruit decay from post-harvest spoilage microorganisms and the relative extension of their shelf life. This strategy consists in the inoculation of viable antagonistic microorganisms (bacteria, yeasts or a mixture of them) or the addition of their antimicrobial products in complete or partial purified form during, at the end or after the chain production [17,18,19]. Biological control based on the use of beneficial microorganisms is receiving increasing recognition, although the number of registered and approved marketed bioproducts containing bioactive yeasts or bacteria remains scarce. Czajkowski and collaborators [20] argue that this situation arises from objective difficulties encountered during registration and marketing, but also from problems of understanding the specific roles of each member of a consortium and their biological activity. Regarding the application of NSYs in bio-protection strategies against grapevine trunk diseases in the vineyard (the fungal pathogen *Botrytis cinerea* causing bunch sour rot), research has increased significantly in recent years [21,22]. In addition, there is a growing interest in the application of this approach on the protection against undesirable microorganisms during the entire wine production chain as well as during the conservation and maturation phases [23,24]. The renewed interest in biological control is due to the growing attention to the use of sulfites in food and particularly in wine. In this regard, NSYs have been proposed as a possible effective natural alternative. Several recent studies have been carried out by using selected strains of *M. pulcherrima* and *T. delbrueckii* at the pre-fermentative stage in the red winemaking process [23,25,26]. Other studies focused attention on the use of other selected strains of *M. pulcherrima* during the cold clarification stage of the Italian Verdicchio white variety underlining the double role of this yeast as a biocontrol agent and wine aroma enhancer [27].

The antimicrobial activity of NSYs has also been investigated in other sectors because the re-evaluation of the use of these yeasts with biocontrol purposes has been extended across the food industry. Some applicative studies also showed their effectiveness in breadmaking or cheesemaking. Valsaraj and colleagues [28] highlighted the biological load of some NSYs and how their toxic killer effects could be used in the food and beverage industry, maintaining food quality and improving the safety in beer, cheese or bread. They suggested that the biocontrol strategy of NSYs isolated from foods and beverages that are naturally fermented may be effective in suppressing wild yeast strains during another fermentation.

No less important is the involvement of the antimicrobial role of yeast strains in the medical field. For over thirty years, Polonelli and Morace [29] reported the killer phenomenon against bacteria, seeing the possibility of using these toxins to counteract the growing phenomenon of antibiotic resistance. In a more recent study, the yeasts *D. hansenii, Pichia fermentans*, *Candida tropicalis* and *Wickerhamomyces anomalus* have been shown to induce bacterial lysis [30]. Chen et al. [31] isolated mycocin-producing strains of *Kluyveromyces marxianus* and demonstrated that crude extracts were effective in preventing *Escherichia coli* disease in mice. In addition, the yeasts *Kluyveromyces lactis* and *K. marxianus*, isolated from cheese, were able to inhibit the growth of pathogenic microorganisms, such as *Listeria monocytogenes* and *Candida albicans* [32]. Finally, *C. albicans* strains isolated from children had inhibitory effects on strains of *E. coli* ATCC 25 922 and *Staphylococcus aureus* ATCC 25 923 [33].

### 2.2. Modalities of Antimicrobial Action of NSYs

The regulation and the control of the growth of undesired microorganisms could be exploited through different mechanisms such as space and nutrient competition, cell-to-cell contact or antagonistic action mediated by antimicrobial compounds production such as mycocins, small peptides, VOCs or extracellular vesicles.

#### 2.2.1. Antagonistic Action: Competition of Nutrients and Space

The main mode of action of yeasts in biocontrol is the competition for space and nutrients [34]. Most organisms can starve pathogens or parasites through iron withdrawal [35]. *Aureobasidium pullulans* possesses a siderophore identified as fusarinin C which has been shown to exhibit antibacterial activity [36,37]. Pulcherriminic acid is a cyclic dipeptide that complexes iron in *M. pulcherrima*. Studies have shown that pigment-deficient mutants of *M. pulcherrima* exhibited reduced or null antifungal activity and iron deprivation of the fungal pathogen. This suggested that the production of this pigment is one of several mechanisms by which this yeast antagonizes plant pathogenic fungi [38]. However, mutants lacking the ability to synthesize pulcherriminic acid still strongly inhibited filamentous fungi, confirming that the antifungal activity was only due to iron deprivation. Therefore, the exact contribution of iron chelators to the yeast biocontrol activity remains to be clarified in detail. Another competition for nutrients was found in *Saccharomycopsis schoenii* in that it is unable to assimilate sulfur, a specific feature of the genus *Saccharomycopsis*. In addition, phytopathogenic fungi and *Trichoderma* species exhibited a similar phenomenon, which may indicate that methionine is an important target for such organisms and has been hotly contested [39]. Pioneering experiments were aimed at evaluating the suitability of an easily transformable *Pichia* (formerly *Ogataea*) *angusta* haploid strain to identify biocontrol-minus mutant clones: while the wild-type strain proved effective in reducing brown rot lesion caused by *Monilinia fructicola* on apple fruit, its derivate leucine-auxotrophic mutant L1 had no significant effect in controlling the pathogen. The addition of exogenous leucine fully restored the biocontrol capability of mutant L1, whereas a leucine stand-alone treatment showed no significant biocontrol effect [40].

Another strategy to compete for space is the formation of biofilm. Biofilms are microbial communities consisting of a single or more species and are considered virulence factors for pathogenic microbes [41,42]. The development of a yeast biofilm begins with the adhesion of single cells to a surface and usually involves cell wall modifications, secretion of an extracellular matrix, and often the formation of hyphae or pseudo-hyphae [43]. In yeasts to be used in biocontrol activities, biofilm formation in wounds is now considered an important mode of action even though the molecular basis of the process and the composition of different biofilms have only been studied in detail for *P. fermentans*. Biofilm formation of *P. fermentans* in apple wounds protects against post-harvest disease, while on peaches it changes from a yeast-like to hyphal growth form and causes rapid decay of inoculated fruit in the absence of a plant pathogen [44]. Biofilm formation has been demonstrated besides that of *S. cerevisiae* also in *P. kudriavzevii*, *W. anomalus* and *M. pulcherrima* [45,46,47,48].

The saprophytic yeast-like fungus *A. pullulans* has been well documented for over 60 years in microbiological literature for its ubiquitarian presence in soil, water, rock surfaces and in both cold and warm areas. A lot of *A. pullulans* strains are known to produce a wide range of natural antimicrobials that are useful for biocontrol applications against plant pathogens mediated by nutrient and space competition and/or VOC production (see Section 2.2.6). *A. pullulans* can be used at both vineyard, for the management of grey mold disease caused by *B. cinerea,* and winery levels, through the synthesis of antifungal compounds, providing a versatile tool for the viticulturist/farmer as well as for the oenologist to combat problems in the field and create a high-quality wine.

#### 2.2.2. Mycocins

Mycocins (killer toxins) are the most investigated yeast antimicrobial compounds. Since Bevan and Makower [49] discovered the killer phenomenon in *S. cerevisiae*, several other yeast species have been found to produce a toxic proteinaceous factor that kills sensitive strains [50,51]. Several potential applications for the killer phenomenon have been suggested in the food industry to control spoilage yeasts in the preservation of food and beverages [52,53].

The mycocin Kpkt produced by *Tetrapisispora phaffii* was first described as an anti-spoilage yeast [52]. Kpkt acts through a specific β-glucanase activity causing irreversible modifications on the cell wall structure and it is codified by the TpBGL2 chromosomal gene [17,54,55]. The recombinant toxin (rKpkt) was recently obtained by transferring the Kpkt coding gene in *Komagataella phaffii* (formerly *Pichia pastoris*) [56]. The recombinant rKpkt, when expressed in *K. phaffii,* displayed a wider spectrum of action than its native yeast [57], reinforcing the idea of the possible application of mycocins in the food and beverages industries. *T. phaffii* was used in mixed fermentation at the pre-fermentative stage to control wild yeasts such as *Hanseniaspora*, *Zygosaccharomyces* and *Saccharomycodes* in the place of sulfur dioxide [11].

Several works have been focused on the study of NSYs able to counteract the development of *Brettanomyces* spp., a relevant dangerous yeast in winemaking [58]. In this context, Pikt and Kwkt mycocins, produced by *W. anomalus* (formerly *Pichia anomala*) and *Kluyveromyces wickerhamii*, respectively [54], are able to counteract *Dekkera/Brettanomyces*. Pikt is an ubiquitin-like protein of about 8 kDa able to interact with the β-1,6-glucan of the cell wall of sensitive yeasts [59], while Kwkt is a protein of about 72 kDa of molecular mass, without any glycosyl residue [11] and β-1,6-glucosidase activity that seems to be involved in the act of blocking the cell cycle function of sensitive yeasts [12]. Cytofluorimetric evaluation showed that both Pikt and Kwkt caused irreversible death of this yeast in a different way from sulfur dioxide that induced a viable but non-cultivable (VBNC) state of *Brettanomyces* with a consequent recovery of yeasts when fresh medium was replaced [12]. Another *P. membranifaciens* strain showed a killer action producing two mycocins denominated PMKT and PMKT2. PMKT binds linear (1→6)-β-d-glucans in the cell wall and Cwp2p plasma membrane receptor of sensitive yeasts, leading to alterations in ionic exchange via plasma membrane [60]. PMKT2, a protein with an apparent molecular mass of 30 kDa, binds mannoproteins and induces cell cycle blockage in the early S-phase of sensitive yeasts and stimulates markers of cellular apoptosis such as the cytochrome c release, DNA strand breaks, metacaspase activation and production of reactive oxygen species at a low dose [61]. The killer activity of *P. membranifaciens* was exploited in winemaking to control the *B. bruxellensis* economic relevant spoilage yeast using mixed fermentation with *S. cerevisiae* and *B. bruxellensis* (inoculum ratio of 1:1). *P. membranifaciens* inhibited *B. bruxellensis* growth without any effects on the fermentation activity of *S. cerevisiae*.

Other mycocins (CpKT1 and CpKT2) active against *B. bruxellensis* are produced by *Candida pyralidae*. Both mycocins were active and stable at pH 3.5–4.5 and with the general conditions of the winemaking environment [62]. Both mycocin CpKT1 and *C. pyralidae* viable yeasts were used in mixed fermentation in red grape juice containing *B. bruxellensis,* determining a decrease in spoilage yeast concentration [63].

Additionally, a strain of *W. anomalus* was proposed as a biocontrol agent against *Brettanomyces*/*Dekkera* spp. [18]. The killer activity of *W. anomalus* is expressed through the release of KTCf20, a mycocin. This mycocin was able to counteract the growth of different spoilage yeasts such as *Brettanomyces/Dekkera*, *Pichia guilliermondii* and *P. membranifaciens*. Moreover, they showed that *W. anomalus* in mixed fermentation did not negatively affect *S. cerevisiae* strains. Another mycocin named WA18 and active against *B. bruxellensis* is produced by an autochthonous *W. anomalus* strain isolated from soil pit, and it exhibited 99% identity with UDP-glycosyltransferase protein [64]. In accordance with de Ullivarri and coworkers [18], the compatibility of this *W. anomalus* strain in mixed fermentation with *S. cerevisiae* yeast was confirmed. Another killer strain belonging to *W. anomalus* has been investigated for its wide potential of antibacterial activity against numerous human pathogenic agents [65]. Interest in these antibacterial mycocins was revealed by Muccilli and Restuccia, [66] who highlighted the potential use against pathogens resistant to conventional antibiotics, such as *Staphylococcus aureus*.

In a recent work, a comparative evaluation of formulates (semi-purified and lyophilized broth) of three well-characterized mycocins was assessed against *B. bruxellensis*. The absence of dangerous effects toward human epithelial cells opens the way for their possible commercial application [67]. Another NSY that shows antimicrobial activity is *T. delbrueckii,* where a mycocin exhibited glucanase and chitinase activities, it is stable in wine environmental conditions and it is active against *B. bruxellensis* and other potential wine spoilage yeasts. Moreover, Ramírez et al. [68] isolated and selected the *T. delbrueckii* strain that produces a mycocin [Kbarr-1]. This toxin is encoded by a dsRNA, TdV-Mbarr-1, which is structurally like M dsRNAs of *S. cerevisiae*, and both seem to be evolutionarily related [69].

All reported studies contributed to demonstrate the global exigence to reduce conventional chemicals by using selected NCYs to ensure high-quality agri-food products with increased aromatic features and/or longer shelf life.

#### 2.2.3. Antimicrobial Peptides (AMPs)

Some peptides produced by yeasts have shown antimicrobial effects against several grape-must/wine-contaminating yeasts. In general, these peptides show lengths of up to 100 amino acids, sorted into variable sequences, and the mode of action involves the disruption of the cell wall in sensitive strains. For example, small peptides with molecular mass below 5 kDa produced by *Candida intermedia* have shown greater antimicrobial specific effects against *B. bruxellensis* [70]. In addition, the antibacterial activity of the same strain against *Escherichia coli*, *L. monocytogenes* and *S. typhimurium* was demonstrated [71]. Similar observations have been reported by Younis et al. [72], where three isolates of *C. intermedia* from raw milk and fruit yoghurt showed antimicrobial activity against *E. coli*, *S. aureus* and *Pseudomonas aeruginosa*.

Another mechanism of interaction in mixed fermentation using NSY is the possible involvement of extracellular vesicles (EVs). In this regard, a recent work on the exo-proteome of EV-enriched fractions in pure and mixed fermentation with six different species of NSYs and *S. cerevisiae* showed a wide diversity of proteins secreted, indicating the presence of interactions and the possible involvement of EVs [73]. The EV-enriched fractions from different species such as *S. cerevisiae*, *T. delbrueckii* and *Lachancea thermotolerans* showed enrichment in glycolytic enzymes and cell-wall-related proteins, particularly the enzyme exo-1,3-β-glucanase. However, this protein was not involved in the here-observed negative impact of the *T. delbrueckii* extracellular fractions on the growth of other yeast species. These findings suggest that EVs may play a role in fungal interactions.

#### 2.2.4. Secreted Enzymes

Some enzymes secreted may be involved in biocontrol action. Indeed, the secretion of enzymes degrading cellular components such as chitinases, glucanases or proteases is a common feature in all kinds of host–pathogen interactions and has been intensively studied. Chitinolytic enzymes allow the degrading of fungal cell walls [74]. Yeasts belonging to genera *Aureobasidium, Candida, Debaryomyces, Metschnikowia, Meyerozyma, Pichia, Saccharomyces, Tilletiopsis, Wickerhamomyces* and *Saccharomycopsis* exhibited this enzymatic activity [16,39]. Chitinases from sources, i.e., fungi and filamentous bacteria, have demonstrated biocontrol activity against plant pathogenic fungi and chitinases are extensively studied as potential biopesticides, targets for resistance breeding or as transgenes in genetically modified plants. Chitinases, probably in an indirect manner, influence biocontrol activity because Chito-oligosaccharides (CHOSs) resulting from chitin degradation are potent inducers of plant immune responses [51].

Glucans are major cell wall components in fungi and exoglucanases are involved in cell wall modification, cell adhesion and resistance to mycocins [75]. A 1,3-β-glucanase from *Candida oleophila* was the first gene cloned in this organism and overexpression or deletion of this gene did not significantly affect *Penicillium digitatum* spore germination, but subsequent studies have documented a reduced inhibitory activity of the β-exoglucanase deletion mutant compared to the wild type and overexpressing strain (in vitro and in fruit), thus demonstrating the involvement of glucanases in the biocontrol activity of yeast [76]. In *W. anomalus*, the deletion of two exo-β-glucanases (PaEXG1 and PaEXG2) significantly reduced the fruit biocontrol activity against *B. cinerea* [77], while the single deletion of PaEXG2 did not reduce biocontrol performance. Exoglucanase activity was also detected in several biocontrol yeasts and was linked to antagonist activity, but without demonstrating a causal involvement.

It has long been proven [78] that in *Rhodotorula glutinis* and *Cryptococcus laurentii*, β-1,3-glucanase activity did not correlate with their respective inhibitory activity against *B. cinerea*. The pathogenicity of yeast species belonging to *Candida, Cryptococcus* or *Malassezia* is related to lipase activity. Several studies have also correlated the role of lipases with the biocontrol action of fungi and bacteria against plate diseases. For this reason, the lipolytic activity of yeasts may represent an aspect to be investigated in relation to biological control [79].

The alkaline serine protease Alp5 of *A. pullulans* reduced spore germination and germ tube length of *Penicillium expansum*, *B. cinerea*, *M. fructicola* and *Alternaria alternata* in vitro and showed a concentration-dependent inhibitory effect on these pathogens on the apple tree [80]. Protease activity has been reported but not confirmed or investigated in the genera *Metschnikowia*, *Pichia* and *Wickerhamomyces*, but has not been further studied or confirmed.

#### 2.2.5. Mycoparasitism

Mycoparasitism is little studied in yeasts, but some studies have shown that *P. guilliermondii* adheres to the hyphae of *B. cinerea* and causes the collapse of the hyphae, presumably due to the secretion of hydrolytic enzymes such as glucanases [81].

Species belonging to the genus *Saccharomycopsis* have been studied against the biocontrol of several clinically relevant *Penicillium* species and yeasts [39].

#### 2.2.6. Volatile Organic Compounds (VOCs)

Volatile organic compounds (VOCs) are the most relevant metabolites that show a biocontrol action. These metabolites are molecules < 300 Da with a low solubility in water and a high vapor pressure and include molecules such as hydrocarbons, alcohols, thioalcohols, aldehydes, ketones, thioesters, cyclohexenes, heterocyclic compounds, phenols and benzene derivatives. Volatiloma is specific for each yeast species as well as the spectrum of action against pathogenic microorganisms [82,83]. VOCs are species-specific and are produced by fungi, bacteria and yeast during their primary and secondary metabolism, limiting the growth of other microorganisms. The chemical composition of VOCs strongly depends on the environment and the pathogen being antagonized. The chemical composition includes alcohols, aldehydes, cyclohexenes, benzene derivatives, heterocyclic compounds, hydrocarbons, ketones, phenols, thioalcohols and thioesters. VOCs are produced by several yeast species to reduce the growth of pathogen molds. In this context, Di Francesco and coworkers [84] have demonstrated both in vitro and in vivo that the VOCs produced by *A. pullulans* reduced the growth and the infection of *B. cinerea*, *Colletotrichum acutatum*, *P. expansum*, *P. digitatum* and *P. italicum*.

Additionally, VOCs formed by NSYs *W. anomalus*, *M. pulcherrima* and *A. pullulans* as well as *S. cerevisiae* showed a biocontrol action against *B. cinerea* on table grape berries [85]. Selected strains of *Cyberlindnera jadinii*, *Candida friedrichii*, *C. intermedia* and *L. thermotolerans* inhibited the formation of both mycelial growth and ochratoxin A in *Aspergillus carbonarius* and *Aspergillus ochraceus* identifying β-phenyl ethanol as the active compound [86,87]. The VOCs of *W. anomalus* prevented spore germination, mycelial growth and toxin production of *Aspergillus flavus* [88]. Similarly, VOCs released by *Candida sake* reduced the incidence of apple rot caused by *P. expansum* and *B. cinerea* [89]. The inhibitory activity of *Sporidiobolus pararoseus* on spore germination and mycelial growth of *B. cinerea* was mainly attributed to 2-ethyl-1-hexanol, whereas *C. intermedia* produced 1,3,5,7-cyclooctatetraene, 3-methyl1-butanol, 2-nonanone and phenylethyl alcohol as the major components of its volatilome during the interaction with this pathogen. VOCs released by *W. anomalus*, *Pichia kluyveri* and *H. uvarum* inhibited *A. ochraceus* growth and ochratoxin A production during the fermentation process of coffee [90].

Recently, Ruiz-Moyano et al. [91] screened yeasts isolated from figs producing antifungal VOCs. A total of 11 out of 34 yeasts, belonging to *A. pullulans*, *Filobasidium oeirense*, *H. uvarum* and *Hanseniaspora opuntiae,* showed a reduction in the growth of *B. cinerea* correlated with the production of 10 volatile compounds: 2 acids (acetic acid and octanoic acid), 7 esters (ethyl propionate, n-propyl acetate, isobutyl acetate, 2-methylbutyl acetate, furfuryl acetate, phenylmethyl acetate, 2-phenylethyl acetate) and a ketone (heptane-2-one).

The *Torulaspora* spp. Strain showed biological action against *Alternaria arborescens*, a causal agent of tomato fruits’ decay. The mechanisms involved were both the production of volatile compounds and the competition for nutrients [92]. In a screening of 147 yeasts and yeast-like fungi, 5 strains belonging to *Anthracocystis* sp., 2 strains of *Aureobasidium* sp. *Rhodotorula* sp. and *Solicococcus keelungensis* produced VOCs and were active against *Aspergillus flavus* that produce aflatoxin B1. Alcohols, alkenes, aromatics, esters and furans, 2-phenyl ethanol and methyl benzene acetate were the most abundant compounds generated by *Aureobasidium* sp. On the other hand, 2-methyl-1-butanol and 3-methyl-1-butanol were significant compounds produced by the other three genera [93]. In another recent work, *Candida pseudolambica* showed a significant reduction of 41.2% of the disease incidence of gray mold in peaches inoculated with *B. cinerea* [94]. Several modalities of action were identified: the growth dynamics, VOCs’ effects, parasitism and biofilm formation. VOCs released from *C. pseudolambica* inhibited the mycelial growth and conidia germination of *B. cinerea*. Fourteen VOCs were identified with the main compounds being 3-methyl-1-butanol and 2-phenylethanol, which made up 85.90% of the relative peak area.

### 2.3. Registered Commercial Products of Biocontrol NSYs Species

Nowadays, there are few products based on yeasts suitable for plant protection that are registered and marketed, such as *C. oleophila*, *A. pullulans*, *Metschnikowia fructicola*, *C. albidus*, and *S. cerevisiae* (details are reported below). However, selected NSYs are also used post-harvest to extend the shelf life of vegetable or in the winery to limit the common use of conventional chemical antimicrobials, such as sulfur dioxide. For example, strains of *M. pulcherrima* and *T. delbrueckii* are commercialized as natural biological control agents with potential to partially or fully replace sulfites. A brief description of the NSY strains used in registered formulations present in the market is reported.

#### 2.3.1. *Candida oleophila*

Species of the genus Candida that strongly inhibit plant pathogens are *Candida diversa* [95], *Candida ernobii* [96], *Candida guillermondii* [97], *C. oleophila* [37], Candida saitoana, *C. sake* [89,98] or *Candida subhashii.* These species are biocontrol agents against molds and post-harvest diseases of hazelnut and citrus fruits.

*C. oleophila* was the first yeast to be developed into a commercial plant protection agent and, regarding the basis for its antifungal activity, several different mechanisms of action have been demonstrated. In addition to competition for nutrients and space, studies of several Candida species have identified hydrolytic enzymes such as proteases, chitinases and glucanases, as well as volatile compounds, which have been implicated in antifungal activity [99].

Furthermore, biofilm formation, high osmotolerance, induction of resistance in the plant/fruit and direct parasitism of hyphae were shown to contribute to the biocontrol activity of *Candida* species [100]. To overcome the inconsistent performance of the initial *Candida*-based biocontrol products, combinations with fungicides, different buffers (e.g., calcium chloride, bicarbonate), chitosan or lysozyme were studied [101]. The *C. oleophila* strains I-182 and strains O have been developed into the biocontrol products Aspire^®®^ and Nexy^®®^, respectively (Ecogen, Langhorne, PA, USA). The latter was the first biocontrol yeast to be registered against a post-harvest disease [100] and *C. oleophila* strain O has been approved as a plant protection agent in Europe in 2015 (European Commission Health & Consumers Directorate-General 2013; European Food Safety Authority, EFSA) [102].

#### 2.3.2. *Aureobasidium pullulans*

The biocontrol activity of *A. pullulans* has been documented in several different strains, but only DSM 14,940 and DSM 14,941 are registered, in admixture Botector-New (Manica), as active ingredients of plant protection products against the disease caused by the bacterium *Erwinia amylovora* and post-harvest diseases (European Food Safety Authority EFSA, 2013) [103]. These two strains of A. pullulans were selected based on their strong inhibition against *E. amylovora*. The two strains were formulated into the Blossom-Protect^®®^ product and tested over several years in different sites under field conditions [104] and also registered for post-harvest apple tree disease control as a Boni-Protect^®®^ product [105]. As with many other yeasts with a biocontrol action, the mode of action of *A. pullulans* involves competition for space and nutrients, but enzymatic activities such as proteases, chitinases or secreted molecules may also be involved (see above). Specific metabolites or enzymes and their contribution to the biocontrol activity of DSM 14,940 and DSM 14,941 have not been identified.

#### 2.3.3. *Metschnikowia* spp.

*M. fructicola* and *M. pulcherrima* are the most studied yeast species regarding biocontrol action. Indeed, they can inhibit a range of post-harvest and plant rot diseases, [85,106,107]. The antifungal activity of *Metschnikowia* species is mediated by a range of mechanisms that involve competition for nutrients (e.g., amino acids, iron), secretion of glucanases and chitinases and the production of volatile organic compounds [38,74,80]. Originally, *M. fructicola* was isolated and discovered in Israel and developed and registered as a biocontrol product to prevent post-harvest diseases, particularly in sweet potato and carrot [108]. *M. fructicola* has also been patented as an antagonist of plant pathogenic microorganisms [109].

#### 2.3.4. *Cryptococcus* spp.

Basidiomycetes belonging to the genus *Cryptococcus* are isolated from water sources, soil and decomposing plant material. *Criptococcus albidus*, *C. laurentii* and *Cryptococcus flavus* strains have been shown to have a post-harvest protective action in peach, cherry, strawberry, tomato, citrus and pome fruit from post-harvest decay [110,111,112]. *C. albidus* was used in the preparation of Yieldplus^®®^ (Anchor Bio-Technologies in South Africa) against *B. cinerea*, *P. expansum* of pome and citrus fruit and *B. cinerea* during post-harvest cold storage of strawberries [113]. The modality of action of this yeast is related to competition for nutrients and space. Moreover, *C. albidus* exhibits glucanase, chitinase and protease activity and produces unidentified volatile compounds that inhibit fungal growth and can display killer activity against *C. glabrata* [114]. However, none of these mechanisms have been directly linked to the inhibitory activity of the target plant pathogens.

#### 2.3.5. *Torulaspora delbrueckii* and *Metschnikowia pulcherrima*

A strain of *M. pulcherrima* registered as LEVEL2 INITIA™ is a biological tool selected with the purpose of preserving the aromatic potential and protecting against oxidative phenomena in the pre-fermentation phases, limiting the use of SO_2_. LEVEL2 GUARDIA™ is another strain of *M. pulcherrima* that is added at very early stages of the production process and colonizes and grows rapidly in the medium, ensuring prompt and efficient bioprotection of red musts. The mode of action is based on the ability to secrete high concentrations of pulcherriminic acid capable of chelating the iron present in the medium, which makes the environment unsuitable for the growth of the contaminating microbiota, facilitating the alcoholic fermentation driven by sequentially inoculated *S. cerevisiae*. A multistarter formulated based on two strains of the species *T. delbrueckii* and *M. pulcherrima* is marketed under the name of ZYMAFLORE^®®^ ÉGIDETDMP and limits the prevalence of unwanted microorganisms on the surface of the harvesting equipment in contact with the grapes.

### 2.4. Application of NSYs in Biopackaging

In a more long-term perspective of the wide application of NSYs, biopackaging has recently led the world of research towards the development and application of natural systems as green packaging in the agri-food industry [115]. Investigations on biopackaging arise from the need to reduce the production and therefore the disposal of plastic and other non-ecofriendly materials. The prefix bio- associated with packaging has a double meaning: biodegradable and/or coming from natural materials. In any case, the objective of food biopackaging is to protect and preserve food products in a sustainable way. Indeed, appropriate packaging has to be capable of maintaining a food’s sensorial and nutritional quality and guaranteeing food safety both in international and national markets for a long time and even in complex conditions [116].

The observation that contaminating microorganisms in food can be controlled during their shelf life by antimicrobial compounds’ release from packaging has led to focused efforts towards active biopackaging systems that incorporate natural antimicrobial compounds (NACs). Castelan and coworkers [117] named these bioactive antimicrobial molecules NACs and reviewed the principal natural sources suitable for this purpose. The NACs are mainly represented by minerals, fruit extracts, essential oils and animal shells but also by some microorganisms or their active molecules. Regarding this last group, there is a growing interest in the development of antimicrobial packaging materials containing natural antimicrobial agents [118]. The same interest has been driven by consumer concerns about health-related issues, such as the use of synthetic antimicrobial agents. Incorporating synthetic antimicrobial agents directly into foods can effectively inhibit the growth and survival of various microorganisms, but consumers demand minimally processed, preservative-free food products with a longer shelf life. Active antimicrobial biopackaging provides a suitable natural alternative to chemical additives as an advanced barrier able to prevent the development of food pathogens.

Since 2012, many articles have appeared regarding the use of antimicrobials of bacterial origin [119] such as bacteriocins synthesized by Archaebacteria, Lactococcus, *Streptococcus* and *Lactobacillus*. For example, the incorporation of nisin and pediocin produced by *Lactococcus lactis* and *Pediococcus acidilactici,* respectively, into polymer matrices was applied in biopackaging, exploiting their bactericidal and bacteriostatic efficacy against *L. monocytogenes* [120,121] for mozzarella preservation.

The effectiveness of yeast in preventing food decay has already been established over the last decades, particularly in post-harvest treatments of fruits to control mold decay [21,122]. Recently, Zhang et al. [16] reviewed the recent literature about the antagonistic role of yeasts in the food industry, highlighting their suitable role among various microbial antagonists. Indeed, yeast and yeast-like fungi as biocontrol tools against pathogens are environmentally friendly, possess greater stress tolerance then bacteria and can potentially be genetically improved. Moreover, the utilization of yeasts is generally considered safe and easily acceptable by the market, and antagonistic yeasts with excellent biocontrol performance have been developed and registered as commercial products.

The incorporation of yeast into the packaging material could impart antimicrobial activity as well as enhancing its nutritional value and serving as the probiotic. For instance, the yeast *Meyerozyma guilliermondii* is non-pathogenic and demonstrates good antimicrobial activity and it is a rich source of vitamins and proteins [123]. In 2013, Coda and colleagues [124] demonstrated the efficacy of *M. guilliermondii* strain LCF1353 for the effective antifungal activity on long-term storage in wheat bread. Therefore, the direct incorporation of *M. guilliermondii* or its metabolites to the packaging material could contribute to increasing the shelf life of food. On this, Atta at al. [125] published a study aimed to develop edible and bioactive food packaging films comprising yeast incorporated into bacterial cellulose in conjunction with carboxymethyl cellulose and glycerol to extend the shelf life of packaged food materials. *M. guilliermondii* strain MT502203.1 and *Gluconacetobacter xylinus* strain ATCC53582 biofilms were developed ex situ then incorporated into the fibrous cellulose matrix. The findings of this study indicate that the developed biofilm could be used as an edible packaging material with high nutritional value and distinctive properties related to the film component, which would provide protection and the extended shelf life of foods.

Recently, Guimaraes and colleagues [126] provided a comprehensive review of the use of edible films and coatings for the incorporation of living microorganisms, aiming at the biopreservation and probiotic capacity of food products. They summarized six benefits and advantages of edible films and coatings containing living microorganisms: physical barrier to protect food, antimicrobial and/or probiotic added value, protection against mechanical damages, increasement of shelf life and green approach.

The ability of two bio-based films of sodium alginate and locust bean gum to deliver the antimicrobial yeast *W. anomalus* cells for the growth control of *P. digitatum* was investigated [127]. The authors confirmed the efficacy of the yeast’s incorporation in preserving the post-harvest quality of artificially infected ‘Valencia’ oranges, where the reduction of green mold was more than 73% after 13 days of shelf life. The edible coating also represents a strategy for conveying probiotic microorganisms and making them readily available in foods which would therefore have an improved nutraceutical value. Tripathi and Giri [128] catalogued about 500 probiotic food products introduced in the market over the last years, mainly yoghurt and other fermented dairy products, such as cheese, ice cream and fermented children’s formula. Because the criteria for a microbial strain to be used as a probiotic includes the ability of remaining viable at high cell count throughout the manufacture and storage of the product, conveying probiotic strains in non-fermented foods through edible coatings could represent a cutting-edge strategy. The first study of an edible coating with yeast dates back to 1994 and concerns the use of *C. oleophila* in cellulose films to extend the storage of table grapefruits [129]. When *C. oleophila* was incorporated in methylcellulose or hydroxypropylcellulose, the storage time was increased by 11 days; moreover, *C. oleophila* was not adversely affected by the incorporation of 0.15% potassium sorbate, the maximum concentration allowed in food products. In the following years, *C. guillermondii* and *Debaryomyces* spp. were applied in formulation to extend the shelf life of oranges. Sharma et al. [130] used chitosan films containing *Candida utilis* for the control of decay in tomatoes caused by *Alternaria alternata* and *Geotrichum candidum*, while Fan et al. [131] tested the incorporation of *C. laurentii* in alginate-based coatings to extend the shelf life of strawberries. These last results showed that the addition of *C. laurentii* had an antagonistic effect and inhibited the growth of molds as well as maintained the overall qualities of the strawberries during prolonged storage. Furthermore, sodium alginate films incorporated with *C. laurentii* did not show any significant effect on color parameters or anthocyanin concentrations in strawberries but were able to maintain fruit firmness during storage. More recently, Yinzhe and Shaoying [132] investigated the effect of carboxymethylcellulose and alginate-based coatings incorporating brewer yeast on grape preservation, highlighting an effective reduced decay compared with the uncoated control and an increase in general quality of the grape because coatings decreased weight loss. Again, Parafati et al. [133] tested the survival and biocontrol ability of *W. anomalus*, *M. pulcherrima* and *A. pullulans* in coated mandarins, where the incorporation of yeasts reduced the incidence of *P. digitatum* in mandarins.

Although many positive effects of active food packaging have been proven by research, there are still few applications. Theoretically, regulation has been well established in the EU, where active packaging is considered as active materials and articles, as they are framed in the Regulation 1935/2004/EC (European Commission, 2004) and Regulation 450/2009/EC (European Commission, 2009); however, biopackaging materials are still limited in commercial applications if compared with traditional packaging applications reported in the literature. The main principle for the successful packaging of fresh and fresh-cut produce is specific, because each produce varies and hence the requirements for packaging and storage vary.

## 3. NSYs as Probiotic Yeasts

Probiotics are defined, by an Expert Panel in 2001, as “live microorganisms which when administered in adequate amounts confer a health benefit on the host”, according to the international organizations FAO (Food and Agriculture Organization of the United Nations, Rome, Italy) and WHO (World Health Organization, Geneva, Switzerland). Since then, many researchers focused the attention on this topic, going from about 750 papers indexed prior to 2001 to more than 20,000 until early 2019 [134]. However, most of the papers use the general term “probiotics” and only a minor part (<3%) use the term probiotic yeasts [135]. Most of the studies conducted so far have been almost exclusively focused on the bacteria population as probiotics and only recently the scenario has opened to yeasts as new probiotics. Effectively, the subject of probiotics and potentially probiotic yeasts has been developing and raising potential for new probiotic products with novel properties, which are not offered by bacteria-based probiotics available on the current market, showing a lot of technological useful traits (Figure 2).

Initial in vitro screening is necessary to propose possible new probiotic candidates. Mainly, the screening needs to include the ability to survive in the host. Indeed, generally probiotics are taken orally reaching the gastrointestinal tract, thus it is important to evaluate their resistance to gastrointestinal conditions (the presence of digestive enzymes, gastric and pancreatic juices, bile salts, pH and body temperature of host). Added to these conditions are the ability to colonize intestinal mucosal surfaces (correlated to auto- and coaggregation capability and surface hydrophobicity), the interaction with the existent gut microbiota (antimicrobial activity toward pathogenic microorganisms) and the exhibition of the antibiotic resistance [136,137]. No less important is also the assessment of the safety of the new candidate against the host: they must not produce toxins, be pathogenic or exhibit hemolytic and DNase activities and gelatinase production [138,139,140]. Lastly, the evaluation of technological features that include the ability of easily cultivation and biomass production and resistance to preservation procedures such as lyophilization, genetic stability and no deterioration are also important [135].

Actually, the only recognized and commercialized yeast for human applications, which fully satisfies the presence of probiotic characteristics, is *S. cerevisiae* var. *boulardii* [141]. According to the current literature and Index Fungrum, 2021, it is closely related to the *S. cerevisiae* wine strains [142]; therefore, it has currently been re-classified as *S. cerevisiae* var. *boulardii*, although it exhibits some unique properties. This yeast is commonly used as biotherapeutic in humans, especially for the treatment of gastrointestinal tract disorders [143] and, more recently, as a starting culture to produce functional and probiotic foods/beverages characterized by health-promoting properties [144].

The recent growing interest for functional and probiotic foods and beverages has prompted scientists to focalize the research on the selection of new strains with both technological and innovative traits such as functional and probiotic properties and bioactive compound production [145,146]. Yeasts are widespread in nature and their relatively easy availability and cultivation make them promising probiotic candidates, especially NSYs, although they are still poorly investigated. However, a recent report from Li et al. [147] described the GRAS status of some NSYs, a necessary prerequisite for their evaluation as probiotics. Generally, these yeasts are isolated from fermented foods and drinks, vegetables, fruit juices, fruits, dairy products, industrial food waste and grains [148,149]. The isolation of these yeasts from human-related food matrices could represent a way to guarantee their safety.

Un-anthropized natural environments and spontaneous processed foods could represent valid sources for the isolation of new strains with probiotic features. A total of 13 out of 180 yeasts isolated from moss on oak, beech tree bark, wine, wineries and grapes, sugar cane juice and papaya leaves revealed the potential probiotic and antimicrobial activity against some pathogenic yeasts. The isolated yeasts belonged to *L. thermotolerans*, *Metschnikowia ziziphicola*, *S. cerevisiae* and *T. delbrueckii* species [150]. Findings highlighted their promising probiotic characteristics, although these aptitudes were strictly strain-dependent [151,152]. Additionally, naturally fermented table olives represented a source of potential probiotic yeasts: five NSYs belonging to *Candida orthopsilosis*, *C. tropicalis*, *D. hansenii*, *P. guillermondii* and *Meyerozyma carribica* showed resistance to 37 °C, pH 2.0, in presence of bile salts and antimicrobial activity against *Salmonella enteritidis* [137].

Several studies described similar properties in other different NSYs including *H. osmophila*, *H. guilliermondii*, *H. uvarum*, *H. osmophila*, *K. thermotolerans*, *K. marxianus*, *P. membranifaciens*, *P. kudriavzevii*, *Pichia masmurika*, *Pichia occidentalis*, *Candida quercitresa*, *C. intermedia*, *C. sake*, *T. delbrueckii*, *W. anomalus* and *M. guillermondii* [141]. They were unaffected by conditions that mimicked a dynamic gastrointestinal system and, thus, could be proposed for new commercial functional foods [153,154]. In a study conducted to compare functional and biotechnological characteristics of potential probiotic yeast strains belonging to *Saccharomyces* and NSYs (genera *Pichia*, *Lachancea, Hanseniaspora*, *Candida* and *Zygosaccharomyces*), it was reported that in aerobic conditions all NSYs showed a higher capacity for adhesion to the Caco-2/TC7 intestine-derived cell line and a higher assimilation for nine prebiotics (cellulose, inulin, melibiose, raffinose, xylan, trehalose, pectin, cellobiose and beta-glucans) than *Saccharomyces* strains. Furthermore, considering other properties, the presence of digestive enzymes, antioxidant activity, antifungal resistance and vitality after sonication treatment found that two strains of *H. osmophila* and one of *L. thermotolerans* were identified as health-promoting probiotics [155]. The evaluation of non-pathogenicity and safe-for-consumption aspects excluded the strain *H. osmophila* 1094 for the high production of biogenic amines, especially tyramine [153].

In 2018, a strain of *Cryptococcus* (95% of ITS sequence similarity with *C. albidus*) from the Red Sea with probiotic traits was firstly described: tolerance to low pH and gastric juice, resistance to bile salts, hydrophobicity, antimicrobial activity and ability to degrade cholesterol [156]. Another NSY with probiotic properties was the strain of wild type *P. pastoris* X-33 that showed the ability to survive the stresses of the gastrointestinal tract and caused no lesions when provided to mice thorough the diet, exhibiting a high antibacterial activity against *S. enterica* serovar *typhimurium* [157]. Mice infected by a virulent strain of *S. typhimurium* showed high survival when supplemented with *P. pastoris* by gavage or via diet. These results concluded that the yeast *P. pastoris* X-33 has probiotic properties with considerable antibacterial activity against *S. typhimurium*. Additionally, *P. kluyveri* was described as a potential probiotic yeast proposed in co-fermentation with *S. cerevisiae* strains and *Lactobacillus paracasei* to obtain a novel, functional fermented maize-based beverage. Similarly, Canonico and coworkers [158] proposed the application of wild non-*Saccharomyces* potential probiotic yeasts to produce a premium craft beer *Kazachstania unispora*, isolated from artisanal sourdough, capable of producing a craft beer with low ethanol content and distinctive aromatic notes. Still, *P. kudriavzevii* DCNa1 and *Wickerhamomyces subpelliculosus* DFNb6 were proposed to ferment cornelian cherry fruit puree with the aim to obtain a functional beverage characterized by low ethanol content, high amounts of alcohols and esters and low levels of aldehydes and alkanes. Moreover, these yeasts remained alive in a dose recommended for a probiotic beverage after 21 days of cold storage and after an in vitro simulated digestion system. They were able to modulate the intestinal microbiota composition, when they were ingested through this beverage, in an in vitro gastrointestinal simulator [159]. However, further studies are necessary to confirm the general probiotic advancement of NSYs with in vivo assay. On the other hand, the probiotic aptitude of NSYs and their use in no- or low-alcohol beverages would be positively considered during functional beverages’ production.

Retracing all the aspects here reported, a general reflection emerges: the long history of yeasts in science, their useful contributions to research and above all human life, as well as the broad prospects for their use in new fields of application lead us to assume that they will continue to accompany scientists for many years, contributing to the improvement of human life, as they have done since the beginning. The recently highlighted potential of NSYs and their specific contribution to biotechnological food matrices represent the central themes of this review.

## 4. Conclusions and Perspectives

In the last two decades, there has been growing interest in NSYs with their improving applications in the agri-food industry with consequent commercial relevance. In addition to the features regarding the quality of food and beverages, as the enhancement of flavor and aroma complexity or ethanol reduction, there is a growing interest in NSYs concerning the antimicrobial issue. In this regard, NSYs constitute a great variety of species that have shown effective antagonistic capacity against several spoilage and pathogenic microorganisms, as attested by this review in which references related to more than 70 yeast species involved in the central topic of biocontrol or probiotic traits were considered (Table 1).

Furthermore, the interest in NSYs, triggered by their rich and diverse reservoir of enzymes and secondary metabolites not typically produced by the conventional *S. cerevisiae* yeast, as well as their metabolic diversity are now fundamental to meet consumer demands for novel sensory properties and health benefits. Effectively, NSYs can be used against pathogenic microorganisms in all production chain steps, starting with primary production in the field, continuing in post-harvest or during the transformation processes and the packaging and storage phases of food and beverages. Additionally, enriched food with the addition of NSYs with probiotic traits represents a safe and easy way to bring fermented foods and beverages with health attributes to the market without increasing costs. Therefore, the consumer would have the possibility of simultaneously consuming fermented food supplemented with a source of viable probiotics, bypassing the purchase of pharmaceutical formulations with high costs and dubious vitality. For these reasons, the metabolic diversity provided by NSYs promises to meet the consumer demands for new sensorial and beneficial health properties.

Moreover, the growing interest in ecological, economical and health sustainability of the agri-food industry in reducing agrochemical treatments but also conventional synthetic additives and antibiotics or antiseptics during technological food transformation represent the starting point to explore this topic. On the other hand, this field of research is still little explored and deserves to be investigated. Little is known about the activities of yeasts and NSYs against very dangerous pathogenic bacteria involved in food processing and preservation. In this way, more in-depth knowledge on the spectrum of action of NSYs towards pathogenic or alterative bacteria together with the study of the mode of action could represent an opportunity to counteract the antibiotic resistance phenomenon, which afflicts not only the medical field but also the food sector. It would therefore be desirable to direct the research towards studies aimed at the interactions between the safe and easily handled NSYs and bacteria, even in complex fermentations involving the interaction of several microbial species, with the aim of clarifying the contribution of the antagonist and the sensitive microorganisms.

Finally, integrated studies of applied microbiology and food technology would be desirable to increase the possibilities of supplying formulations in which the microorganisms remain alive for long time and in which the management of the inoculum and fermentation process is as flexible, economical and easy as possible.

## Figures and Tables

**Figure 1 microorganisms-11-01450-f001:**
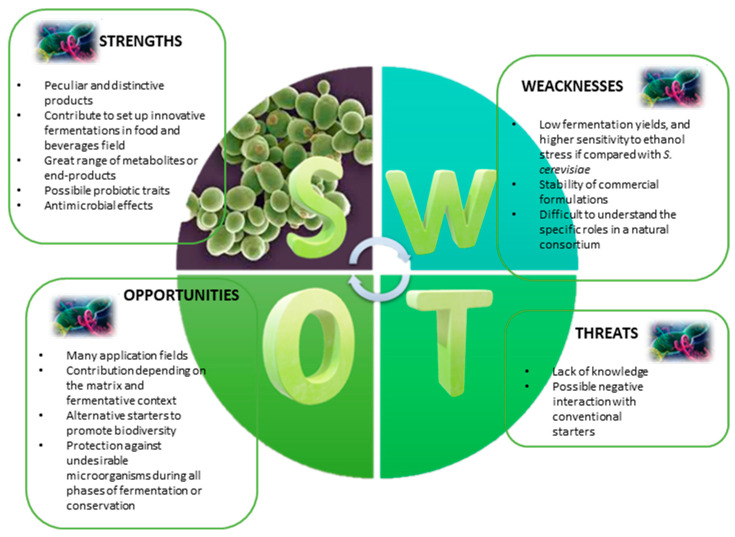
Identification of key strengths (S), weaknesses (W), opportunities (O) and threats (T) related to the use of non-*Saccharomyces* yeasts (NSYs) in fermented foods and beverages.

**Figure 2 microorganisms-11-01450-f002:**
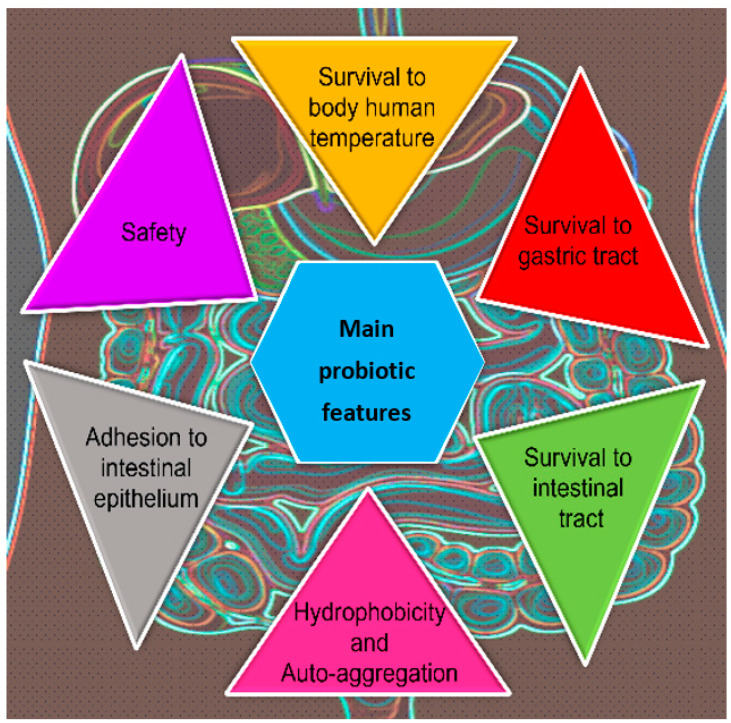
Graphical representation of the main probiotic technological characteristics.

**Table 1 microorganisms-11-01450-t001:** List of all NSYs reported in the review and relative matrices, specific activity and reference.

NSYs	Products	Distinctive Antimicrobial Features	References
*Aureobasidium pullulans*	Guava fruit	Antibacterial activity mediated by fusarinin C	[37]
*Aureobasidium pullulans*	Citrus	Antimicrobial biofilm formation	[48]
*Aureobasidium pullulans*	Plants	Biocontrol mediated by nutrient, space or VOC	[18,52,54,55,56]
*Aureobasidium pullulans*	Mandarins’ edible coating	Anti-mold activity	[80]
*Aureobasidium pullulans*	Table grape berries	Biocontrol VOC mediated	[80]
*Aureobasidium pullulans*	Fruits	Biocontrol VOC mediated against *A. flavus*	[86]
*Candida* spp.	Top, hazel and citrus fruits	Antagonist of filamentous fungi	[104]
*Candida albicans*	Medical field	Inhibitory effect on *S. aureus* and *E. coli*	[33]
*Candida guillermondii*	Oranges’ biopackaging	Anti-mold activity	[128]
*Candida inetrmedia*	Milk	Probiotic activity	[139]
*Candida intermedia*	Wine environment	Anti-spoilage yeasts (wide spectrum) AMPs mediated	[69]
*Candida intermedia*	Medical field	Antimicrobial against *E. coli*, *S. aureus* AMPs mediated	[70,71]
*Candida laurentii*	Strawberries’ edible coating	Antimold activity	[129]
*Candida oleophyla*	Fruits	Antimold activity (*P. digitatum*) EVs mediated	[75]
*Candida oleophyla*	Table grapefuits’ biopackaging	Antimold activity	[127]
*Candida orthopsilosis*	Table olives	Probiotic activity	[135]
*Candida pseudolambica*	Peach	Biocontrol VOC mediated against *B. cinerea*	[92]
*Candida pyralidale*	Wine	Anti-*Brettanomyces* activity mycocin mediated	[63]
*Candida quercitresa*	Indigenous fermented food	Probiotic activity	[139]
*Candida sake*	Apple rot	Biocontrol VOC mediated against *P. expansum* and *B. cinerea*	[87]
*Candida sake*	Milk products	Probiotic activity	[139]
*Candida tropicalis*	Medical field	Bacterial lysis	[30]
*Candida tropicalis*	Table olives	Probiotic activity	[135]
*Candida utilis*	Tomatoes biopackaging	Anti-mold activity	[128]
*Debaryomyces hansenii*	Medical field	Bacterial lysis	[30]
*Debaryomyces hansenii*	Table olives	Probiotic activity	[135]
*Hanseniaspora guillermondii*	N.D. ^1^	Probiotic activity	[139]
*Hanseniaspora uvarum*	Coffee	Biocontrol VOC mediated against *A. ochraceus*	[88]
*Hanseniaspora uvarum*	N.D. ^1^	Probiotic activity	[139]
*Kluyveromyces lactis*	Medical field	*L*. *monocytogenes* and *C. albicans* growth inhibition	[32]
*Kluyveromyces marxianus*	Medical field	*E. coli* diseases prevention in mice	[31]
*Kluyveromyces marxianus*	Medical field	*L. monocytogenes* and *C. albicans* growth inhibition	[32]

^1^ N.D.: Not Determined.

## Data Availability

Data are contained within the article.

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
