# Peer review of "Biocontrol and Probiotic Function of Non-Saccharomyces Yeasts: New Insights in Agri-Food Industry"

_microorganisms, 2023, doi:10.3390/microorganisms11061450_

Round 1

Reviewer 1 Report

Line 62 add citation

“could be advantageous in the bakery  industry, providing greater diversity than S. cerevisiae-based products, and may be useful in reducing or avoiding yeast intolerance” sentence extracted from Zotta et al 2022. It is suggested to modify

Line 396 a 403 It is suggested to add scientific papers or trade names to find out what is available in the market

Line 724 but also food sector change “but also agri-food secto”. The agricultural sector also suffers from bacterial resistance in crops

Author Response

Detailed response to reviewers

1° reviewer

Line 62 add citation

 Answer: Citation added

“could be advantageous in the bakery industry, providing greater diversity than S. cerevisiae-based products, and may be useful in reducing or avoiding yeast intolerance” sentence extracted from Zotta et al 2022. It is suggested to modify

Answer:  modified the sentence

Line 396 a 403 It is suggested to add scientific papers or trade names to find out what is available in the market

Answer:  we added  at this point the sentence “details reported below” since the commercial products with NSY were detailed in the paragraphs  dedicated for each species.

Line 724 but also food sector change“but also agri-food secto”. The agricultural sector also suffers from bacterial resistance in crops

Answer: correction made

Reviewer 2 Report

Congratulation for a great manuscript!

I hope that my suggestion can contribute to its already high quality.

Agri-food industry is a plausible expression. I would suggest to include a short definition in introduction, because it would orient even better the reader what to expect.

Also, I suggest to put a bit more emphasis in the text as well, that what is the “agri-food” in those examples bring up.

Lines 253 to 266: I do not see clearly the agri-food connection here. You just write “…in mixed fermentation…”.

Lines 287 to 296: What kind of pure and mixed fermentation happened here?

I have similar questions regarding the probiotic non-Saccharomyces yeasts. Please, see my remark regarding that section later.

I have found the overall structure of the article very good. However, I’m not sure that the section 2.4 is at the right place. Perhaps switching it with Section 2.3 would be a good idea. Section 2.3 is about real/industrial application of biological control, while the examples you show about biopackaging still seems to be one step before that (I mean everyday application).

One more suggestion to Chapter 2.3: I believe it would be more exact information for the readers if you define “formulations”. What kind of products are them? How should the reader imagine?

And a remark about the topic of Chapter 2.4. I have found it really interesting and informative, but you haven’t mentioned in the earlier parts of the article that you will deal with biopackaging. So it was a kind of surprise (although nice) to read this topic. It is worth to mention this and perhaps the formulation as well either in the abstract or the introduction.

Probiotic yeast

I have found the introductory part of this chapter a bit lengthy, but I can accept if you do not shorten it, because it is put together well, and the topics it includes are actually necessary.

The examples you bring show that research on the application of probiotic yeasts – especially NSY – is quite fresh. Research is mostly about studying properties and determining whether they are probiotic.

You bring only a few examples when product was made with NSY – all of them are alcoholic beverages (from Line 666). I suggest to make a note here that as far as I know food or beverage that contains alcohol cannot be considered as functional.

Table 1 gives a really good summary and overview. Please, consider to include it and the text that precedes the Table 1 (Lines 680 to 691) in Chapter 4: Conclusion and perspectives or in an independent chapter.

The English of the manuscript is good, but sometimes more complicated than necessary. I had problem with the very first sentence of the Introduction chapter. Please, double-check it, because I think something is missing from it.

There are only a few typos in text, please, correct them:

Line 47: specie-specific should be species-specific

Line 89: revaluation probably should be re-evaluation

Line 205: graph 2.1.6 probably should be 2.2.6

Line: 319: this citation cannot be found in the References, and here in the text only the number should be mentioned

Line 488: I would suggest to have the abbreviation of "natural antimicrobial compound" in parenthesis right after the expression.

Starting from Chapter 2.3: none of the microorganism name is in Italics.

The English of the manuscript is good, but sometimes wording of the sentences is more coplicated that necessary. I have found very few typos, and I mentioned them in my comments for the Authors.

Author Response

 2° reviewer

Congratulation for a great manuscript!

I hope that my suggestion can contribute to its already high quality.

Answer: thank you for  your appreciation of the manuscript

Agri-food industry is a plausible expression. I would suggest to include a short definition in introduction, because it would orient even better the reader what to expect.

Answer: as suggested we added a short idefinition in introduction

Also, I suggest to put a bit more emphasis in the text as well, that what is the “agri-food” in those examples bring up.

Answer: following you suggestion we added “details are reported below”  in introduction of the paragraph  since more details in the formulations/products with NSY were reported.

Lines 253 to 266: I do not see clearly the agri-food connection here. You just write “…in mixed fermentation…”.

Answer: as suggested, we enclose the reason to used the killer strain P. membranifaciens

Lines 287 to 296: What kind of pure and mixed fermentation happened here?

I have similar questions regarding the probiotic non-Saccharomyces yeasts. Please, see my remark regarding that section later.

Answer: we added “using NSY” to better explain the effect of small peptides by NSY toward  spoilage microorganisms

I have found the overall structure of the article very good. However, I’m not sure that the section 2.4 is at the right place. Perhaps switching it with Section 2.3 would be a good idea. Section 2.3 is about real/industrial application of biological control, while the examples you show about biopackaging still seems to be one step before that (I mean everyday application).

 Answer: we revised numeration of paragraphs  and enclose  a phase to indicate the different steps for real application between biocontrol  at different stages of process and biopackaging  a modality of biocontrol application  (modifications at the start of paragraph 2.4)

One more suggestion to Chapter 2.3: I believe it would be more exact information for the readers if you define “formulations”. What kind of products are them? How should the reader imagine?

 Answer:  we change formulation with product . Ther commercial product of NSY could be differ for the  additives and conservation compounds.

And a remark about the topic of Chapter 2.4. I have found it really interesting and informative, but you haven’t mentioned in the earlier parts of the article that you will deal with biopackaging. So it was a kind of surprise (although nice) to read this topic. It is worth to mention this and perhaps the formulation as well either in the abstract or the introduction.

 Answer. We added a mention biopackaging in the abstract

Probiotic yeast

I have found the introductory part of this chapter a bit lengthy, but I can accept if you do not shorten it, because it is put together well, and the topics it includes are actually necessary.

The examples you bring show that research on the application of probiotic yeasts – especially NSY – is quite fresh. Research is mostly about studying properties and determining whether they are probiotic.

Answer.: we added a phase with the indication of the necessity to confirm with in vivo studies  the probiotic characters of NSY line 690-692

You bring only a few examples when product was made with NSY – all of them are alcoholic beverages (from Line 666). I suggest to make a note here that as far as I know food or beverage that contains alcohol cannot be considered as functional.

 Answer.: we added a phase with the indication of the necessity to confirm with in vivo studies  the probiotic characters of NSY line 690-692

Table 1 gives a really good summary and overview. Please, consider to include it and the text that precedes the Table 1 (Lines 680 to 691) in Chapter 4: Conclusion and perspectives or in an independent chapter.

 Answer: We enclose Tab.  In the chapter 4 after a brief introduction

The English of the manuscript is good, but sometimes more complicated than necessary. I had problem with the very first sentence of the Introduction chapter. Please, double-check it, because I think something is missing from it.
Answer: we revised all manuscript for the mistakes

There are only a few typos in text, please, correct them:

Line 47: specie-specific should be species-specific

correct

Line 89: revaluation probably should be re-evaluation

correct

Line 205: graph 2.1.6 probably should be 2.2.6

correct

Line: 319: this citation cannot be found in the References, and here in the text only the number should be mentioned

correct

Line 488: I would suggest to have the abbreviation of "natural antimicrobial compound" in parenthesis right after the expression.

correct

Starting from Chapter 2.3: none of the microorganism name is in Italics.

 Correct and revised all manuscript for mistakes